# Use of Pelleted Diets in Commercially Farmed Decapods during Juvenile Stages: A Review

**DOI:** 10.3390/ani11061761

**Published:** 2021-06-12

**Authors:** Mohd Amran Aaqillah-Amr, Ariffin Hidir, Mohamad N. Azra, Abdul Rahim Ahmad-Ideris, Muyassar H. Abualreesh, Mat Noordin Noordiyana, Mhd Ikhwanuddin

**Affiliations:** 1Higher Institution Centre of Excellence (HICoE), Institute of Tropical Aquaculture and Fisheries, Universiti Malaysia Terengganu, Kuala Terengganu 21030, Terengganu, Malaysia; aaqillahamr_92@yahoo.com (M.A.A.-A.); hidirariffin@gmail.com (A.H.); a.ideris@umt.edu.my (A.R.A.-I.); 2Institute of Tropical Biodiversity and Sustainable Development, Universiti Malaysia Terengganu, Kuala Terengganu 21030, Terengganu, Malaysia; azramn@umt.edu.my; 3Department of Marine Biology, Faculty of Marine Sciences, King Abdulaziz University, Jeddah 21589, Saudi Arabia; mabulreesh1@kau.edu.sa; 4Faculty of Fisheries and Food Sciences, Universiti Malaysia Terengganu, Kuala Terengganu 21030, Terengganu, Malaysia; diyananoordin@umt.edu.my; 5STU-UMT Joint Shellfish Research Laboratory, Shantou University, Shantou 515063, China

**Keywords:** feed, feeding diets, macro-micronutrients, feeding behavior, pellet–decapod performances

## Abstract

**Simple Summary:**

Information on the diet composition, food types, and feeding behavior of decapods is important for developing well-formulated diets in aquaculture, since feed constitutes the largest operational cost in crustacean hatcheries. The use of formulated feed for decapods at a commercial scale is still in the early stages. This is probably because of the unique feeding behavior that decapods possess: being robust, slow feeders and bottom dwellers, their feeding preferences change during the transition from pelagic larvae to benthic juveniles as their digestive systems develop and become more complex. This article presents an overview of recent progress on the nutrition and feed formulation of commercially farmed decapods during the juvenile stages. In this review, we attempt to update the information on the topic from the last 25 years and examine challenges and opportunities in the development of formulated diets, considering diet composition and decapod feeding behavior during the juvenile stages, which is vital for developing a better quality of feed formulation in hatcheries.

**Abstract:**

The increasing market demand for decapods has led to a considerable interest in cultivating decapod species at a larger scale. Following the development of hatchery technologies, most research has focused on the development of formulated feeds for commercially farmed decapods once they enter the juvenile stages. The use of formulated feed for decapods at a commercial scale is still in the early stages. This is probably because of the unique feeding behavior that decapods possess: being robust, slow feeders and bottom dwellers, their feeding preferences change during the transition from pelagic larvae to benthic juveniles as their digestive systems develop and become more complex. The current practice of decapod aquaculture involves the provision of juveniles with food such as natural diet, live feed, and formulated feed. Knowledge of nutrient requirements enables diets to be better formulated. By manipulating the levels of proteins and lipids, a formulated feed can be expected to lead to optimal growth in decapods. At the same time, the pellet’s physical characteristics are important factors to be considered upon formulating commercially farmed decapod feeds, considering the unique feeding behavior of the decapod. However, most published studies on decapod nutrition lack data on the physical characteristics of the feed types. Thus, it is difficult to establish a standard feed formulation that focuses on the physical pellet properties. Moreover, careful consideration must be given to the feeding behavior of species, as decapods are known as bottom feeders and are robust in terms of handling feed. Information on the pellet forms, diet composition, and unique feeding behaviors in commercially farmed decapods is gathered to suggest potential better formulated diets that can optimize growth and reproduction. Thus, the purpose of this review is to summarize the information that has been published to date and to come up with suggestions on ways to improve the feed formulation in decapods that comply with their feeding behavior and nutrient requirements. Further research is needed to explore the potential of the pelleted feed at the adult stage so the decapod can take full advantage of the nutrients present in the pellets.

## 1. Historical Developments in Cultivation and Feed Formulation of Decapods

Decapods are valuable sources of aquatic food protein, and their fisheries and aquaculture support the economic growth of many coastal countries [1]. The increasing demand for seafood products has led to considerable interest in cultivating decapod species at a larger scale. The cultivation of decapods in various countries began during the 1980s with raising juveniles from the wild. In 2018, aquaculture reported a strong growth in decapod production, primarily of penaeid shrimp, crabs, and spiny lobsters (9.4 million tons), as compared with the previous year [2]. Decapod aquaculture occurs in different countries of Southeast Asia from India to the Philippines, including Malaysia, Thailand, and Indonesia. In the western hemisphere, the exploration of intensive freshwater prawns began in the 1960s, including in Europe and the USA [3]. In the early 20th century, several Western countries developed a large-scale production of the larval stages of American and European lobsters (*Homarus americanus* and *H. gammarus*) in an attempt to restock natural populations. At the same time, decapod aquaculture in Western Australia gained popularity as a result of the difficulty of harvesting wild stocks. According to the Department of Primary Industries and Regional Development, Western Australia’s aquaculture production has some of the finest, most sought after decapods and other seafood in the world, with a value of exports of $595 million in 2015–2016, and rock lobster accounting for 76% of the state’s fisheries exports, with a value of $453 million. In Malaysia, the development of decapod aquaculture is relatively recent, beginning with marine shrimp culture in the mid-1930s [4]. For mud crabs, the commercialized aquaculture program for mud crab in Malaysia began in 1991 and eventually became one of the main components of local fisheries [5]. Both penaeid shrimp and portunid crabs are important candidates for decapod culture in Indo-Pacific countries such as Malaysia, Indonesia, Australia, India, Bangladesh, Philippines, Japan, and China.

The success of decapod farming is dependent on the variety of diets [6,7,8]. Current practices of commercially decapod farming involve the provision of juveniles with food such as natural diet, live feed, and formulated feed [9]. The provision of decapods with these types of feeds might be termed exogenous feeding, which is defined as feed that is provided by farm practitioners to caged decapods. Following the development of hatchery technology, most studies have focused on the development of formulated feeds for decapods. The commencement of formulated feed to decapod seeds begins once they enter juvenile stages, which is common practice for whiteleg shrimp or Pacific white shrimp (*Litopenaeus vannamei*)*,* portunid crabs (*Scylla* sp. and *Portunus* sp.), and giant river prawns or giant freshwaters prawns (*Macrobrachium rosenbergii*) [10]. By manipulating the levels of proteins and lipids, a formulated feed can provide optimum growth in decapods.

The developments of a formulated feed for decapods begins with the use of fish oil (FO) and fishmeal (FM) as the main sources of lipids and proteins, with other ingredients such as wheat flour being the main source of carbohydrates (CHO). The inclusion of vitamins and minerals, probiotics, and other feed additives, when combined, satisfy the growth demand. Previously, forage fish have been the basic ingredient for both FM and FO due to their high protein and lipid levels as well as other micronutrients such as fatty acids and amino acids. The expeditious rise in the aquaculture industry has placed a significant amount of pressure on forage fish stocks to fulfil the demand for formulated feed [11]. Thus, as the development of this formulated feed has progressed, FO and FM have been reduced and other alternative ingredients such as terrestrial animal sources and vegetable oils have been used instead.

Current research into the development of decapod formulated feeds is geared towards the juvenile stage, but limited information is available on decapod groups in the adult stage. This is probably because of the unique feeding behavior that decapods possess: being robust [12], slow feeders [13] and bottom dwellers [14]. In addition, most published studies on commercially farmed decapod nutrition lack data on the physical characteristics of the feeds, such as water stability, palatability, and digestibility. Due to these issues, it is difficult to establish a standard feed formulation that focuses on physical pellet properties. In this review, we attempt to update the information and examine the challenges and opportunities of the development of formulated diets, considering diet composition and decapod feeding behavior, which are vital for developing a better quality of feed formulation in hatcheries. This review endeavors to analyze the existing information on the topic from the last 25 years using ‘feed formulation aquaculture’ and ‘feed formulation crustacean’ as keywords in the Web of Science Core Collection database (https://apps.webofknowledge.com accessed on 25 June 2020) to come up with suggestions on how to improve the feed formulation in decapods to suit their feeding behavior and nutrient requirements.

## 2. Decapod Feeding Biology

The decapod’s life cycle begins with an egg or embryo, which is fertilized into zoea/larval stages. Decapods are grouped into different class sizes based on their morphology and a morphometric approach: juveniles, sub-adults (immature), and adults (mature); these class sizes are distinct between decapod groups and are species-specific. Decapod’s larvae undergo a series of molts before becoming postlarvae, then juveniles. They develop into an immature/sub-adult decapod before reaching the adult/mature form, at which time, they are ready to reproduce. Decapods typically have two pairs of appendages (antennules and antennae) in front of the mouth and paired appendages near the mouth that function as jaws, which affects their feeding selection. Many decapod crustaceans are described as bottom feeders and scavengers that feed on dead animals that reside on the seafloor [15]. Most decapods are nocturnal—only active at night in search of food and staying in their burrows during the day. Several decapod species, such as green mud crabs (*Scylla serrata*) [16], blue pearl crayfish (*Cherax albidus*) [17], and tropical spiny lobsters (*Panulirus ornatus*) [18], are active predators that prey on smaller creatures using their claws. They are also cannibalistic and are very aggressive towards each other. These species have a pair of strong claws and pincers that will be used for feeding and fighting. Individuals that are killed during a fight will be eaten by other crustaceans.

In addition, several species are restricted to certain environments that affect the feeding selection between species and between life stages [19,20,21,22]. For example, mangrove crabs or mud crabs (*Scylla* spp.), which live in inundated sea levels, are opportunistic omnivores that prey on slow-moving mollusks, carrion, and plant materials, as well as cannibalizing other crabs [23]. In contrast, blue swimming crabs (*Portunus pelagicus*), which live in the open sea, are elite carnivores and scavengers that feed on teleost fish and invertebrates including mollusks and crustaceans [24]. Other decapod species, such as lobsters from the family Palinuridae, inhabit various habitats from shallow intertidal surf zones to ocean depths characterized by perpetual darkness and soft mud-ooze substrates, which greatly affects their feeding selection. For example, the scalloped spiny lobster *(**Panulirus homarus rubellus*), which inhabits the surf zone, prefers to eat bivalve mollusks such as mussels (*Perna perna*) [25], whereas the Caribbean spiny lobster (*Panulirus argus*), which resides on the seafloor and frequently inhabits burrows and rock crevices, preys on benthic species, including mollusks, echinoderms, and crustaceans [26].

Moreover, feeding preferences also change at different growth stages, for example, the pelagic larvae of many decapod groups such as shrimp and crabs are generally opportunistic, preying on anything suspended in the water, such as plankton (phyto- and zooplankton) [27]. These feeding preferences will change during the transition from pelagic larvae to benthic juveniles whose digestive systems are more complex [28]. Whiteleg shrimp are described as omnivorous during the postlarval stages, ingesting both zooplankton and phytoplankton [29]. On the other hand, mud crabs are elite carnivores during the larval stages, feeding on zooplankton such as rotifers, copepods, and brine shrimp (*Artemia*) [30]. Similar findings were observed in Caribbean spiny lobster larvae (phyllosoma), which are categorized as carnivorous from their first feeding [31]. These commercially farmed decapod species typically change their feeding strategies as they grow to adult stages, from primarily carnivore to omnivore and opportunistic feeding. Table 1 describes decapod feeding behavior during the juvenile stages.

### 2.1. Factors That Affect Feeding of Decapods

#### 2.1.1. Biotic Factors

Biotic factors that affect feeding selection in decapods involve the sensory basis, which includes vision, chemoreception, mechanoreception, and electrosensory systems. In adult decapods such as prawns, shrimps, and crabs, vision is not as important as the other sensory systems since they are nocturnal [15,32,33]. At the same time, other decapods such as the tropical spiny lobster use chemoreception to locate food from the beginning of the juvenile stage since this species resides on the seafloor [18]. The role of vision can be significant during the juvenile stage as these decapods are diurnal and visually detect suspended feed particles during daytime feeding [10]. Meanwhile, chemoreception is described as a process whereby decapods respond to chemical stimuli or chemoattractants in their environment, which depends primarily on the senses of taste (gustation) and smell (olfactory) [34,35]. The chemoattraction of the formulated feed influences the palatability. In the case of a feed formulation that uses alternative protein sources from plant-based materials such as soybean meal (SBM) and canola meal, it is important to include feed attractants.

On the other hand, mechanoreception is defined as the ability of a decapod to detect and respond to mechanical stimuli such as touch, sound, and changes in pressure or posture in their surrounding environment. In decapods, mechanoreception is used to avoid predators or detect prey. The detection of feeding takes place when the larvae encounter food particles (i.e., plankton), usually through tactile reception, whereby the larvae will integrate visual and proprioceptive information as a frame before recognizing them as prey or potential feed [36]. At the same time, electrosensory processes involve the decapod antennae, which typically bear chemoreceptors and mechanoreceptors [37]. The feeding selection in decapods using this electrosensory system usually takes place in situations of poor visibility, such as decapods with ablated eyestalks. Studies performed on ablated juveniles of whiteleg shrimp to determine the role of sensory appendages in feeding showed that the shrimp seized a pellet using the maxillipeds from currents generated by the swimming legs and then ingested it, although it did not visually see the pellet [10]. Similar observations have been recorded in crabs, where ablated grapsid crabs (*Pachygrapsus transversus*) used their maxillipeds as food detection and handling tools [38].

#### 2.1.2. Abiotic Factors

Abiotic factors such as light and day length, temperature, water quality, and the physical properties of the food greatly affect decapod feeding responses. The presence of light is especially important in the decapod during larval stages because, compared with adult decapods, they are primarily nocturnal during the mature stage [39]. Similarly, water temperature can also be a major driving effect against feeding selection. Decapods are known as poikilothermic: their body temperature closely follows changes in water temperature and their metabolic rates are typically temperature-dependent. Feeding trials on red swamp crayfish (*Procambarus clarkii*) and white river crayfish (*P. zonangulus*) showed different responses in feeding rates, where the optimum temperature was 32 °C for both species, with the red swamp crayfish consuming significantly more feed than the white river crayfish [40]. This difference signifies the response of the feeding rate to differences in geographical distribution.

Meanwhile, water quality directly affects feeding responses in decapods. Decapod species depend on their chemical senses for foraging and social interactions, so a low water quality may result in a low feeding rate. High turbidity and total suspended levels in a closed environment will increase the water temperatures, reducing the water clarity and dissolved oxygen, and thereby reducing the feeding intake in decapods. Furthermore, the physical properties of food, such as palatability, particularly attracts decapods, thus affecting their feeding activity. Animal proteins that release certain amino acids generally attract decapods [41,42], although, in decapods, visual cues tend to predominate.

**Table 1 animals-11-01761-t001:** Decapod’s feeding behaviour during the juvenile stages.

Decapod Group	Species	Feeding Behavior			Reference
		Feeding Habit	Feeding Rate	Feeding Time	
Shrimp	*Litopenaeus vannamei*	Ingestion of food is controlled by the mouthparts and esophageal chemoreceptors	Up to satiation	Shrimps were fed once daily in the late afternoon	Derby et al. (2016) [43]
	*Palaemonetes varians;* *P. elegans*	Shrimps are slow and continuous eaters	10% body weight	Fed once a day in the morning (10:00)	Palma et al. (2008) [44]
	*Penaeus vannamei*	N/A	5% of cumulative shrimp body weight per tank	Once per day in the morning	Park et al. (1995) [45]
	*Litopenaeus vannamei*	N/A	Fed in excess	Shrimps were fed four times per day	Galkanda-Arachchige et al. (2019) [46]
	*Penaeus vannamei*	N/A	Fed initiating with 10% of total shrimp biomass, adjusting it weekly	Fed to satiation three times a day (08:00, 13:00, and 18:00)	Gil-Núñez et al. (2020) [47]
	*Litopenaeus vannamei*	N/A	7% average body weight	Fed twice daily (08:00 and 17:00)	Simião et al. (2019) [48]
Crayfish	*Cherax quadricarinatus*	Slow feeding response	Fed to excess	Fed three times daily (07:30, 12:30, and 16:00)	Thompson et al. (2003) [13]
	*Cherax albidus*	Slow feeder/manipulates food using the mouth appendages before ingestion	5% body weight	Fed every day between 08:00 and 09:00	Volpe et al. (2012) [17]
	*Cherax tenuimanus*	Slow intake, prolonged handling, long intervals between food intakes	6.5% body weight	Fed daily	Jussila and Evans (1998) [49]
Crab	*Scylla serrata*	Show preference for detritus	Fed to satiation (about 2–3.5% body weight)	Fed twice daily at satiation level	Catacutan (2002) [50]
	*Scylla paramamosain*	N/A	Fed with excess diets	Fed twice a day at 08:00 and 18:00	Zhao et al. (2016) [51]
	*Scylla serrata*	N/A	6% body weight	Fed twice daily (at 07:00 and 17:00)	Unnikrishnan et al. (2010) [52]; Unnikrishnan and Paulraj (2010) [53]
Lobster	*Panulirus ornatus*	Cannibalistic; relies on chemoreception; slow feeder	50% body weight	Fed twice daily (morning and afternoon)	Marchese et al. (2019) [18]
	*Panulirus argus*	Opportunistic predator	Lobster fed at a rate below the level required to reach satiation (2% of lobster wet weight)	Fed twice daily	Perera et al. (2005) [54]
Prawn	*Macrobrachium rosenbergii*	N/A	The experimental prawns were fed twice daily (07:00 and 18:00 h)	10% of the wet body weight/day	Kangpanich et al. (2017) [55]

N/A: Not available.

## 3. Nutritional Requirements of Juvenile Stages

In decapod feedings, protein, lipid, and carbohydrate (CHO) are described as the most important components of the nutrient classes, acting as the main sources of nutrients for embryonic development and growth [56]. Table 2 shows the macro- and micronutrients of different decapod groups during the juvenile stages.

### 3.1. Protein

Protein is one of the macronutrients in decapod feed that plays a role in promoting growth, fattening, and reproductive processes. Protein encompasses amino acids as a primary source of essential nutrients required for growth. The dietary protein requirement in adult decapods is higher for carnivorous species compared to herbivorous species [57]. Optimum protein levels are especially important in juvenile decapods since they grow through molting activities. An inadequate amount of protein hinders growth [56], sometimes causing mortalities, especially in juveniles during the prolonged intermolt period [53]. However, a dietary protein surplus results in water deterioration from the degradation of protein leftovers, which form ammonia or total ammonia nitrogen [58].

Information on species-specific dietary protein requirements is of vital importance to ensure good growth and maturation. Since protein is the most expensive component of the diets, many studies have explored the use of low-protein sources as the main energy source, sparing protein for growth [57]. Investigations have been carried out to determine the protein requirements of different commercially farmed decapods such as prawns, shrimps, and crabs during the juvenile stage. For instance, an analysis of the growth performance of whiteleg shrimp during the juvenile stage revealed that the optimum requirement for protein is 34.5% [59]. For juvenile mud crabs, high growth performance was recorded when the crabs were fed with diets containing 32–40% protein, while reduced growth was noticed when juveniles were fed higher levels of protein [50]. A later experiment by Unnikrishnan and Pulraj [53] mentioned that 46.9–47.03% crude protein led to the best growth performance in juvenile mud crabs, while a high mortality rate was seen when they were fed with 15% protein.

Apart from protein, the amino acids and micronutrients must also satisfy decapod’s needs in order to lead to good growth performance. The requirements for essential amino acids such as histidine, isoleucine, lysine, leucine, methionine, phenylalanine, threonine, tryptophan, and valine have been established for a number of species such as the kuruma prawn (*Marsupenaeus japonicas*) [60], sesarmid crab (*Episesarma singaporense*) [61], river prawn **(***Macrobrachium americanum*) [62], European lobster [63], and red swamp crayfish [64]. Methionine and lysine are most used in the feeding of commercially farmed decapods due to their wide availability, but are limited in plant and rendered animal byproducts [65]. The requirement for methionine in decapods is primarily to improve growth performance and feed conversion. It is recorded that the dietary requirement for methionine in whiteleg shrimp and giant tiger prawns (*Penaeus monodon*) was 0.45% [66] and 2.4% [67], respectively. Meanwhile, an accepted dietary lysine requirement range of 1.6–2.1% was recorded for the diet of cultured shrimp [65].

Other than growth factors, amino acids predominate in feed and act as stimulants that affect the behavioral response in decapods through chemical stimuli [68]. This involves the mechanism of leaching amino acids from the feed during pellet immersion, which can be identified by the decapod through chemoreceptors. Tests on the inclusion of squid, crustacean and krill meal, fish and krill hydrolysates, and betaine product as feed effectors in the diet of giant tiger prawns revealed that the prawns showed a significant preference for feeds containing crustacean meal or krill meal when more amino acid leachates were observed in the two attractants, which initiated the shrimp responses towards the feeds [42]. Similarly, experiments on whiteleg shrimp showed a higher affinity for a diet containing feed attractants with complex amino acids at a 1.0% inclusion level, particularly because of the presence of complex amino acids such as alanine, valine, glycine, proline, serine, histidine, glutamic acid, tyrosine, and betaine [69]. Meanwhile, a growth trial on tropical spiny lobsters showed that the inclusion of 10% krill meal in the diet formulation enhanced feeding and growth performance due to the presence of amino acids such as glycine and taurine, which were released in higher quantities [18]. Holme et al. [57] suggested that the incorporation of purified amino acids into decapod diets may improve growth performance through their qualities as attractants and stimulants of feed intake.

### 3.2. Lipids

Lipids encompass various classes of organic molecules such as triacylglycerols, phospholipids, sterols, waxes, carotenoids, and fatty acids (FA) [70]. Studies have demonstrated that lipid levels in most decapods increase with size and growth stages. Some reserved lipids are catabolized as energy sources, while others are stored in the gonad for structural purposes, such as maturation and eicosanoid synthesis [71]. Information on lipid requirements is very important in the development of formulated feeds to ensure that the nutrients suffice for good growth and maturation [72]. Collective prior studies on decapods have concluded that optimum growth for decapods at the juvenile stage can be achieved with a total lipid level from 7.5–8% in red claw crayfish (*Cherax quadricarinatus*) [73,74], 2–10% in yellow mud crabs (*Scylla paramamosain*) [51], 10–12% in whiteleg shrimp [75,76], and 9% lipid (with 31% crude protein) in spiny lobsters (*Jasus edwardsii*) [77].

Meanwhile, FA are known for their function as precursors of crustacean hormones and play an important role in the regulation of cell membranes [78]. Highly unsaturated fatty acids (HUFA) are responsible for survival, maintaining high growth and reproductive rates, as well as high food conversion rates in both marine and freshwater organisms [79]. Arachidonic acid (ARA), eicosapentaenoic acid (EPA), and docosahexaenoic acid (DHA) are examples of HUFA-derived omega-6 (n-6) and omega-3 (n-3) FA typically found in marine fish sources; they originate from the phytoplankton and seaweed that are part of their food chain [80,81]. Several studies have demonstrated that DHA, EPA, and ARA play important roles in the survival and growth of decapod larvae, where a deficit of HUFAs may result in poor growth, lower survival, and prolonged intermolt periods [82].

The dietary requirements of n-3 HUFA for the optimum growth performance of several decapod species during their juvenile stages are 0.86% for whiteleg shrimp [83]; 2.01% and 1.27% at 7% and 12% dietary lipids, respectively, for yellow mud crabs [79]; and 3% in 7.5% lipids for giant tiger prawn [84]. Meanwhile, the overall consumption of diets containing ARA, EPA, and DHA with an optimal n-3:n-6 ratio helps to optimize decapod growth. Kangpanich et al. [55] recorded no significant differences in the growth performance of giant river prawns fed with diets containing either FO or alternative oil sources from marine algae (*Schizochytrium* sp.) and soybean oil, even though diets with FO had a higher n3:n6 ratio than the alternative oil (0.96 and 0.54, respectively), suggesting the alternative oil as an effective substitute for FO. Most decapod species obtain their dietary requirements for HUFA from the diets, as they do not have the ability to synthesize them de novo, except for their limited capacity to convert one form of PUFA to another form through elongation and desaturation [80,85]. ARA can be obtained through a series of steps of elongation and desaturation of linoleic acid (LA), whereas both EPA and DHA can be obtained from the elongation and desaturation of α-linolenic acid (ALA) [86]. This limitation is why these fatty acids are considered essential [87]. During the desaturation and elongation process, both n-3 and n-6 PUFA from ALA and LA will compete for the same desaturation enzyme to produce HUFA, particularly ARA, EPA, and DHA, which satisfy the HUFA requirements [88]. However, studies on the metabolism of these PUFAs is limited in decapod species, probably because of a lack of desaturase enzymes [89].

### 3.3. Carbohydrates

Carbohydrates (CHO) are known as a primary energy source and have the same importance in the diet as proteins and lipids. Unused dietary CHO during metabolism will be accumulated in the hepatopancreas in the form of lipids and glycogen (GLY) [90,91]. When entering a period of starvation, CHO will be the first to be depleted, followed by lipids and protein [57,91]. The dietary CHO utilization in decapods is usually correlated to the glucose transport efficiency as the final product of CHO digestion [92], which is regulated by the crustacean hyperglycemic hormone (CHH) [93]. CHH is actively involved in molting and reproduction [94], osmoregulation [95], and the metabolism of fatty acids and CHO [96].

CHO aids in the molting of most decapod species, where a high molting rate can be observed during the larval and juvenile stages. The GLY stored in the hepatopancreas will be used as a precursor of chitin synthesis, which assists in the molting activities of decapods [97]. Various concentration of GLY are observed throughout the molt cycle. The GLY reserved for molting activities in decapods is discussed by Galindo et al. [97]. During the premolt period, an increased concentration of GLY is recorded in the hepatopancreas (2.75–3.75 mg/g), which will be used for the formation of a new cuticle during ecdysis, and a decreasing level was noted during the late premolt to early postmolt stage (3.5–3.13 mg/g). Meanwhile, during the late postmolt period, increased feeding activity was observed in decapods, and an increasing hepatopancreatic GLY (4.25 mg/g) reserve was seen. Information on dietary CHO requirements, as well as the patterns of GLY reserves in the hepatopancreas throughout the molt cycle, is very helpful for shortening the intermolt period in juvenile decapods so that they grow faster.

The requirement of dietary CHO for optimum digestion and metabolism was recorded in the range of 20–30% of the total diet in Caribbean spiny lobsters [98], 24.08% in yellow mud crabs [99], and 18% in oriental river prawns [100]. In principle, the CHO levels required for optimal growth for decapods are 20–30% of the total diet [91]. The dietary requirement for CHO in formulated feed can be obtained from wheat flour. Besides providing decapods with energy from CHO, wheat starch also provides pellet integrity and stability, which agglutinates feed particles through starch gelatinization [101]. Since CHO is rather cheap, the use of CHO as a direct energy source can be advantageous in terms of growth and profitability. Thus, they are widely used as an alternative to lipids as an energy source that contributes to protein-sparing effects during growth [102]. Reducing the protein source ingredients in the diet formulation led to higher dietary requirements for CHO sources in juvenile European lobsters, which opens up opportunities for the production of sustainable and economically viable formulated feeds that reduce the inclusion of protein in the diets [63]. However, higher total CHO consumption can adversely affect decapod performance, leading to slow growth and a high mortality rate, as CHO’s ability to spare dietary proteins is limited [103]. Guo et al. [104] observed a high mortality rate in juvenile whiteleg shrimp after a four-week feeding trial where shrimp were fed with 25–35% corn starch as a source of dietary CHO, and concluded that the optimum corn starch level may be 10–20% when the diets contain 38% protein, 5.3–5.4% lipids, 8.26–8.31% ash, and 12.9–21.5% CHO. Feeding experiments on juvenile European lobsters revealed that individuals fed with 31% and 23% CHO showed the lowest metabolic rates given a protein level of 40% in the formulation [63].

In response to various stressors, decapods will effectively regulate their glucose concentration as directed by CHH; this situation is termed hyperglycemia [105]. When faced with these stressors, decapods require additional energy, which is regulated by CHH [106]. For instance, hemolymph glucose level in the decapod will decrease during starvation periods. The release of CHH stimulates the hydrolysis or breakdown of GLY in the muscle and hepatopancreas, resulting in increases in the glucose concentration in the hemolymph. During a prolonged period of starvation, GLY stored in the hepatopancreas and muscles will decrease due to supplying the body with glucose through glycogenolysis; a critical point is reached where crabs will become increasingly sedentary as a strategy to conserve their limited remaining energy to survive.

**Table 2 animals-11-01761-t002:** Macro and micronutrients in feed formulation of decapods during juvenile stages.

Decapod Group	Macronutrients						Micronutrients		Feed Additives	Reference
	Protein	Carbohydrates	Lipid Derivatives				Vitamin	Mineral		
			Lipid	Cholesterol	Fatty Acids	Carotenoid				
Prawn	47.3%	N/A	7.5%	0.5%	3.0% EFA	Carophyll pink: 0.15%	1.6%	2.0%	Ethoxyquin, squid mantle muscle, L-a-phosphatidylcholine, crystalline amino acids, sodium alginate, tetra-sodium-pyrophosphatem, α-cholestane, α- cellulose	Glencross et al. (2002) [84]
	Isonitrogenous feed 39%	30.8–32.50%	10.15–10.48%	N/A	n-3/n-6: 0.54–0.65	N/A	1.0%	1.0%	Shrimp shell meal, corn grain	Kangpanich et al. (2017) [55]
	39.18%	35.47%	6.91%	N/A	n-3/n-6: 0.69 EPA/DHA: 0.81	N/A	1.0%	2.5%	Soybean lecithin, choline chloride, cellulose, squid paste, calcium phosphate, beer yeast cell, spray dried blood powder	Li et al. (2020) [107]
Shrimp	Isonitrogenous feed 21% dry weight	N/A	77.1–85.9%	3%	N/A	N/A	2.5%	2.0%	Soy lecithin, antifungic, antioxidant (ethoxyquin), Vitamin E	Martínez-Rocha et al. (2012) [108]
	30%	42.1%	6%	0.5%	N/A	N/A	1.0%	4.7%	Lecithin, alpha cellulose, alginate, sodium hexametaphosphate	Velasco et al. (1998) [109]
	35%	N/A	8%	0.2%	DHA: 0.5% ARA: 0.13%	N/A	2.0%	0.5%	Calcium phosphate dibasic, lecithin, StayC	Samocha et al. (2010) [110]
	32.1%	48.1%	5.84%	N/A	N/A	N/A	8.53%	8.53%	Soybean lecithin, alginic acid	Gonzalez-Galaviz et al. (2020) [111]
	40.08–42.93%	33.09–36.4%	7.37–8.39%	0.1%	N/A	N/A	0.5%	0.2%	Lecithin, alginate	Suresh et al. (2011) [41]
	34.2% to 36.3% dry weight	40.5% to 44.3%	3.9% to 6.0% dry weight	N/A	N/A	N/A	1.8%	0.5%	Choline chloride, Stay-C 35% active	Galkanda-Arachchige et al. (2019) [46]
	36%	N/A	8%	0.1%	N/A	N/A	1.8%	0.5%	Choline chloride, Stay-C250 mg/kg, CaP-diebasic, lecithin, chromium oxide	Fang et al. (2016) [112]
	42.2%	N/A	9.1%	0.5%	N/A	N/A	2.0%	2.0%	Calcium phosphate, soya lecithin	Palma et al. (2008) [44]
	39.7%	30.7%	9.45%	0.16%	N/A	N/A	0.28%	0.28%	Krill meal, monocalcium phosphate, lecithin	Derby et al. (2016) [43]
	34.8% protein in feed with soy meal and 29.3% protein in feeds with FM	38.76% in feed with soy meal and 22.45% in feed with FM	6.65% in feed with soy meal and 5.84% in feeds with FM	N/A	N/A	N/A	0.93% in feed with soy meal and 0.85% in feed with FM	0.93% in feed with soy meal and 0.85% in feed with FM	Soy lecithin, alginic acid, cellulose, antioxidant	Gil-Núñez et al. (2020) [47]
	35.8% to 36.6% dry weight	34.7% to 38.9%	7.9% to 8.1%	0.2%	N/A	N/A	0.5%	0.5%	Lecithin-soy, methionine, lysine, titanium dioxide	Weiss et al. (2019) [113]
	Isonitrogenous feed 40% dry weight	N/A	Isolipidic feed 9.00% dry weight	0.02%	N/A	N/A	1.2%	1.0%	Lecithin powder 97%, amygluten	Moniruzzaman et al. (2019) [114]
	Isonitrogenous feed 35% dry weight	31.93–32.78%	8.18–8.63% lipid	N/A	ARA:1.68%; EPA: 2.87%; DHA: 4.66%	N/A	15%	25%	Dicalcium phosphate, antifungal, antioxidant, lysine, methionine, garlic powder	Tazikeh et al. (2019) [115]
	Isonitrogenous feed 36% crude protein	N/A	7.9–9.00% lipid	0.11%	N/A	N/A	0.25%	0.25%	Antioxidant, antifungic agent, Vitamin C, choline chloride,	Gamboa-Delgado et al. (2019) [116]
	37%	38.32 to 38.88%	10%	0.5%	N/A	1.46% (5% from 29.23% carotenoid extracted)	1.0%	1.0%	Monocalcium phosphate, cellulose	Simião et al. (2019) [48]
Crayfish	Isonitrogenous with 39.02% to 39.74% dry weight	41.38% to 44.00% dry weight	Isolipidic 7.03% to 7.53% dry weight	12.6% to 12.9% dry weight	Saturated with 2.52% to 2.72% dry weight and unsaturated with 4.51% to 4.81% dry weight	N/A	N/A	Sodium (1.4% to 1.5%), Calcium (3.3%) & Iron (0.7% to 1.3%)	N/A	Volpe et al. (2012) [17]
	Isonitrogenous (40% protein as-fed basis)	28.33%	7.03%	0%	ARA: 1.09% EPA: 3.58% DHA: 7.94%	N/A	2.0%	0.5%	Lecithin, dicalcium phosphate, Vitamin C, choline chloride	Thompson et al. (2003) [13]
Crab	44.85% to 46.73% dry matter	N/A	7% and 12% lipid	0.50%	DHA/EPA ratio between 2.2 and 1.2 at 7% and 12% lipid, respectively	N/A	1.00%	1.50%	Monocalcium phosphate, choline chloride, cellulose	Wang et al. (2021) [79]
	Isonitrogenous with 43.64 to 46.08% dry weight	17.2 kJ g^−1^	Dietary lipid level of 8.52–11.63% (op timum 9.5%)	0.8%	ARA: 0.5%; EPA: 6.9%; DHA: 6.1%	N/A	3.00%	2.00%	Lecithin, sodium alga acid, squid paste, cellulose	Zhao et al. (2015) [117]
	Isonitrogenous feed with 45% crude protein	N/A	Isolipidic diets containing 9.5% oil (FO, lard, safflower oil, perilla seed oil or mixture oil	0.8%	ARA: 0.5%; EPA: 14.1%; DHA: 11.7%	N/A	3.00%	2.00%	Lecithin, sodium alga acid, squid paste, cellulose	Zhao et al. (2016) [51]
	46.9% to 47.03% dry weight	N/A	Isolipidic feed ~8% dry weight	0.50%	N/A	0.009% β-carotene	1.50%	5.00%	Cellulose, dextrin, lecithin	Unnikrishnan and Paulraj (2010) [53]
	Isonitrogenous with 45% dry weight	N/A	Isolipidic with 10.8% dry weight	0.50%	0.13% ARA; 0.64–0.66% EPA & 0.37–0.38% DHA	0.009% β-carotene	1.50%	5.00%	Cellulose, dextrin, lecithin	Unnikrishnan et al. (2010) [52]
	32 to 40% dry weight	17.2 MJ kg^−1^	6% or 12% dry weight	0.1%	N/A	N/A	1.50%	0.50%	Seaweed, soy lecithin, dicalphos	Catacutan (2002) [50]
	Isonitrogenous 48.5%	N/A	5.3 to 13.8% lipid dry weight	1.0%	0.36–0.4% ARA; 6.54–7.03% EPA; 2.29–2.81%	0.01% Astaxanthin	4.00%	4.00%	Taurine, choline chloride, vitamin A, Vitamin D_3_, Vitamin E	Sheen and Wu (1999) [118]
	46.6% protein dry weight	N/A	8.6% lipid dry weight	0.51%	N/A	0.01% Astaxanthin	4.00%	4.00%	Taurine, choline chloride, vitamin A, Vitamin D_3_, Vitamin E	Sheen (2000) [119]
	44.0–45.7% dry weight	N/A	1.1% to 1.08% lipid dry weight	0.5% dry weight	0.2% ALA, 0.2% ARA, 0.2% DHA dry weight	0.01% Astaxanthin	4.00%	4.00%	Taurine, choline chloride, vitamin A, Vitamin D_3_, Vitamin E	Sheen and Wu (2002) [120]
**Lobster**	Isonitrogenous 53% dry weight	N/A	10.04%	2%	N/A	1% Carophyll pin (8% astaxanthin)	1.1%	0.6%	Lecithin, Stay-C	Marchese et al. (2019) [18]
	25% and 35% protein	23.75–24.73%	6.2–7%	N/A	N/A	N/A	5%	5%	Vitamin C, Vitamin E, Calcium carbonate, dicalcium phosphate	Perera et al. (2005) [54]

N/A: Not available. EFA: Essential Fatty Acid.

## 4. Development of Formulated Feed for Juvenile Decapod

### 4.1. Type of Formulated Feed

There are two main types of feed processing technology that have been introduced in aquaculture: the extruded (pressured) pellet and the steam pellet. The extrusion technique involves the use of a feed extruder, whereby pellets are forced through a die using higher pressure and steam heat before being left to cool and having a vitamin and mineral premix added. The extrusion method is different from the steam pellet in that the extruder does not use any pellet binder to add adhesion to the particles [121], where they only expand through gelatinization of starch [122]. Tuber starches such as potato and tapioca are popularly used as binding agents since they are high in amylopectin–amylose [123] and starch enzyme amylase [124]. These starches become activated and absorb large volumes of water during the gelatinization process. Once cooked, the starch containing amylose leaches from the granules, which increases the viscosity in the dough, aiding with the thickening of feed during formulation [123]. The gelatinization of starch helps to improve feed digestibility in decapods [125]. For this reason, the use of extruder feed is better than a steam pellet as it offers high stability and functional properties [124]. Meanwhile, one can prepare a steam pellet comprising a mixture of all the feed ingredients with a certain volume of water added to make a dough; this is cooked for a certain period of time and left to cool before the addition of a vitamin and mineral premix [121]. The dough will later be shaped according to preference and heated constantly in an oven until a constant weight is achieved. This technique is more cost-effective compared with the extrusion technique. However, the steam pellets have lower stability compared to the extruded pellets. There are several pellet forms for decapods used in the hatchery for commercial purposes: dry pellet, semi-moist pellet, and moist pellet. These feeds can be differentiated in terms of moisture content, where the moisture level for dry pellets is generally less than 10% [46], 62–80% for moist pellets [126,127], and 35% for semi-moist pellets [128]. Generally, wet, moist, and semi-moist diets are more effective in terms of promoting high growth and feed efficiency owing to their soft texture and palatability. High moisture or humidity levels in formulated feeds leave them more prone to deterioration from mold growth after a long storage. Excess moisture content in the formulated feed will support the growth of bacteria, yeast, and mold, which deteriorates the feed quality. These microbes generally grow faster in a feed medium that contains high water activity (aW), which refers to the excess water in the feed. aW is defined as the ratio between the vapor pressure of the feed and the vapor pressure of water under a completely undisturbed balance with the surrounding air media. The aW parameter is often used in bacterial culture media, where microbes cannot multiply when the aW is below 0.900 [129]. Meanwhile, aW in the pellets defines better protection against bacterial and mold growth, where lower aW in the pellets are preferable [130]. The optimum aW level in the formulated feeds has been standardized to 0.65 as the limit for the safe storage of foods [131]. In decapod aquafeed (i.e., shrimp pellets), the common aW is 0.5043, as reported by Carter et al. [132]. In this review, two basic types were considered for intensive farming: dry pellets and moist pellets (semi-moist will be included as they fall under the same category as moist pellets).

#### 4.1.1. Dry Pellet

Dry pellets can be used in a variety of forms: dry-sinking pellet, extruded sinking pellet, and extruded floating pellet. Suitable feed ingredient selection, together with proper manufacturing procedures such as an extrusion or steaming process, ensures high-water stability pellets, which is the main criterion for producing high-quality feeds. Overall, dry-sinking pellets are more practical for bottom feeders [133] such as shrimp [134], prawns [121], lobsters [135], crayfish [13,17], and mud crabs [16]. Necessary for the creation of water-stable dry pellets are good binding agents and finely ground ingredients to ensure the maximum adhesion of the binder molecules.

#### 4.1.2. Moist Pellet

Moist, or wet, pellets are soft pellets consisting of a combination of high-moisture ingredients and dry pulverized ingredients. The use of moist pellets led to high growth performance in juvenile rock lobsters (Jasus edwardsii) [127], freshwater crayfish [136], and green mud crabs [120]. Although the use of moist pellets is widely accepted among decapods, it is highly desirable to have the advantage of storage without the need for a refrigerator in order to prevent fungal growth and mold problems. This has led to the innovation of semi-moist pellets, which have been successfully developed at a laboratory scale. Compared to moist pellets, the moisture content of semi-moist pellets is lower, and under the permissible level to avoid yeast and mold growth, with the addition of chemical agents [137].

### 4.2. Pellet Characteristics Requirement

Table 3 summarizes the decapod’s pellet forms in relation to pellet and decapod performances. The success of decapod farming has highlighted the importance of physical pellet characteristics, which directly emphasizes the significance of artificial or formulated diets to replace live and fresh foods. The success of formulated feed may be controlled by the moisture content in the diet, which directly affects the physical forms. The high moisture content in the pellets is often associated with nutrient leaching since it dissociates easily upon entering the water. Apparently, the low pellet stability and durability resulting from high moisture content may not be suitable for decapods, partly because some species are aggressive in handling food [138]. In addition, the proper storage and handling of feed products may be difficult to achieve, as is the case with wet pellets. Since wet pellets have a high moisture content, rapid spoilage, such as from mold problems, is unavoidable during long storage periods [139]. Other physical pellet attributes, such as the palatability, type of binder, water stability, and durability, as well as buoyancy, are important to avoid pellet disintegration from decapods’ strong mastication and from long exposure to water.

#### 4.2.1. Palatability and Attractability

Palatability is defined as the acceptance of the feed by decapods, whereas attractability involves the decapod’s orientation towards the presence of any feed that has been offered [41]. Optimization of feed intake is determined by the good physical attributes of the pellets, which includes their palatability and acceptability to decapods, taking into consideration species behavior and physiological requirements as well [137]. The attractability of the diet helps the decapod to locate the feed faster due to the strong chemical cues, thus reducing the duration of pellets’ immersion [127]. Insufficient levels of attractants can result in low feed intake and eventually in the poor growth of the organisms.

Both palatability and attractability have become primary factors in the development of cost-effective feed since decapods have great senses of smell, taste, and sight geared towards the search for food. Diets of low palatability and attractability may result in decapods not being able to meet their nutritional needs. Essentially, the measurement of feed palatability and attractability can be determined following Derby et al. [34] by carrying out sequential feeding of pellets to an individual within a certain period of time (i.e., 60 min) until it does not consume the last pellet. During the experiment, pellets will be delivered to each species individually, one pellet at a time. This process will be repeated for additional pellets until the decapod does not consume the last pellet within the feeding duration. The time taken for the decapod to grab the pellet will be recorded as the time from when the decapod put the food in its mouth until it is finished eating, recorded as feeding rate (total number of pellets eaten per total time spent, mg/min).

#### 4.2.2. Water Stability and Durability

Pellet water stability (WS) is defined as the ability of the pellet to retain both its integrity and nutrients while in water, until it is consumed by a decapod [134]. Meanwhile, durability is defined as the ability of the pellet to maintain its shape during handling, transportation, and inflation upon transmission to water without breaking into smaller particles [140]. A higher WS in pellets defines its effectiveness in optimizing feed consumption in decapods during harsh handling and vigorous mastication of the feed so that the nutrients required for growth will be fulfilled [17]. External factors such as high water currents and strong aeration in the tank will all tend to accelerate pellet disintegration, resulting in nutrient leaching [135,136] and, consequently, increased water clarity and turbidity from the suspended materials. The pellet WS and durability are critical for adult decapods such as crabs, lobsters, and crayfish, where larger pellet sizes are used [136], with longer soaking time and the least possible leaching of nutrients [134]. The use of binders helps to hold feed components together, minimizing void spaces and maintaining pellet integrity, thus producing a more compact and durable pellet [136]. It is suggested that pellets must maintain a minimum of 90% dry matter retention after a 1-h exposure in water [139]. A study carried out by Sudaryono [141] revealed that a WS experiment on the dry pellets for juvenile tiger prawns had a low dry matter weight loss (<10%) and that the shrimp showed higher apparent dry matter digestibility and apparent protein digestibility. Similar observations were noted in juvenile blue pearl crayfish, where the pellets showed a stable decrease in WS after 24 h of immersion [17].

The determination of pellet WS was carried out by soaking an individual pellet with a known weight in a beaker containing water for 30, 60, 120, 240, and 360 min [142]. After the required immersion time, the pellets were recovered, while the remaining pellet dissociates in water were filtered through filter paper and then freeze dried for 24 h until constant weight was achieved. The mean differences between the pellets before and after immersion were used to calculate the percentage of dry matter loss (leaching). At the same time, the freeze-dried pellets may be subjected to nutrient (protein and lipid) leaching through a proximate analysis.

Meanwhile, the durability of the formulated feed can be determined using several laboratory methods such as the tumbling box method, Holmen durability tester, and Stokes hardness tester. The durability test using the tumbling box method can be carried out following the method from Adedeji et al. [143]. Five hundred grams of pellets are placed into two separate compartments of custom-made tumbling equipment. One compartment has five metal parts (nuts) to simulate a harsher handling environment, while the other does not. Before emptying the compartment, the cup maker is run for 10 min. The crushed sample will be sieved through a U.S. Standard Sieve No. 270 (0.053 mm) to separate the unbroken feed, and then reweighed. The particle durability index (PDI) is calculated as the percentage of the mass of surviving particles to the total mass of the particles.

#### 4.2.3. Type of Binder

Many types of binders have been used while formulating high-durability pellets to increase WS and minimize nutrient leaching through the added cohesion of the particles and the reduction of void spaces. Binders used include agar, starch, gelatin, carrageenan, and carboxymethylcellulose (CMC). Practically, binders that can be digested and assimilated are chosen. Polysaccharides such as starch play an important role in decapod feed, providing necessary CHO as well as being a binder responsible for the adhesion of the feed components [101]. Starches such as maize, millet, guinea corn, wheat, and cassava improve pellet durability, contain high protein levels, and make a good binder for extruded feed pellets [144]. These types of binder are capable of generating air traps in formulated feeds, thus improving the physical integrity of the feeds in water. On the other hand, unbranched polysaccharides from seaweed, such as agar, sodium alginate, and carrageenan, have been widely applied in the field of aquaculture nutrition, mainly as binders [145]. Investigations on two types of binders (lignosol and agar) and binding methods (microbinding and microcoating) in the diet suggested that the use of either lignosol or agar at 2% levels added to the diet by the microbinding method [44] led to higher growth performance in both juvenile Atlantic ditch shrimp (*Palaemonetes varians*) and rockpool prawns (*Palaemon elegans*). Ruscoe et al. [136] compared the use of carrageenan, CMC, agar, and gelatin as binders at different concentrations in freshwater crayfish and concluded that 5% concentrations of carrageenan and CMC were significantly better than agar and gelatin. Meanwhile, research carried out by Paolucci et al. [146] regarded agar as performing better when compared to both sodium alginate and carrageenan during feed manufacturing. Agar is usually activated when heated up to 80–85 ℃, and the binding of feed components generally begins once the solution cools down to a gelling temperature of 32–43 ℃ [146].

#### 4.2.4. Buoyancy

Floating feed is especially important during the decapod larvae phase to ensure optimum feed intake since the larvae live at the water surface [147]. Meanwhile, long-term sinking pellets are preferable in juvenile and adult stages for their less expanded structure and high densities, which are more suitable since both juvenile and adult decapods such as freshwater crayfish [135], shrimp [148], and mud crabs [16,149] are described as bottom feeders and slow eaters. The determination of pellet buoyancy will be based on the sinking velocity, for which the time taken for the feed to sink to the bottom will be recorded [150].

**Table 3 animals-11-01761-t003:** Decapod’s pellet forms with pellet and decapod performances.

Decapod Group	Type of Feed(s)				Pellet Performance		Decapod Performance				Reference
	Pellet Forms	Shape/Size	Moist/Dry	Binding Agent	Leaching/Stability	Acceptability/Palatability	Digestibility/Energy	Growth Performance (GP)	Protein Efficiency Ratio (PER)	Feed Conversion Ratio (FCR)	
Prawn	Extruded pellet	<155 mm strand	Dry pellet	Wheat gluten, starch, sodium alginate	High water stability was observed in each of the experimental diets	All diets were readily accepted by prawns	N/A	Prawns fed with a diet containing 7.5% lipid with 3% EFA had the highest weight increase, but was not significant with prawns fed with 7.5% lipid with 4.3% EFA	N/A	N/A	Glencross et al. (2002) [84]
	Steam pellet	N/A	Dry pellet	α-starch binder	N/A	N/A	N/A	Prawns fed alternative oil with 2% *Schizochytrium* sp. and 1% soybean oil had the highest weight gain	N/A	Prawns fed alternative oil with 3% *Schizochytrium* sp. had the highest FCR	Kangpanich et al. (2017) [55]
	Extruded pellet	1.5 mm diameter	Dry pellet	Flour	N/A	N/A	Excessive dietary lipid (>7.91%) showed a lack of protein-sparing effect on growth and nutrient utilization levels	Prawns fed 6.91% lipid had the highest GP	Prawns fed 6.91% lipid had the highest PER	FCR was highest in prawns fed 8.89% lipid	Li et al., 2020 [107]
Shrimp	Extruded pellet	2 mm	Dry pellet	Wheat starch, whole wheat	N/A	All diets readily consumed with no indication of feed rejection	N/A	Shrimp fed without HUFA supplements showed reduction in growth	N/A	FCR values of shrimp with HUFA supplements were similar with commercial diets	Samocha et al. (2010) [110]
	Extruded pellet	N/A	Dry pellet	Wheat meal, corn starch	N/A	N/A	N/A	Fast growth (FG) shrimp genotype had a higher growth rate than the high resistance (HR) shrimp genotype	N/A	Both FG and HR showed no feed efficiency differentiation (fed an animal- or vegetable-based diet)	Gonzalez-Galavis et al. (2020) [111]
	Extruded pellet	2.4 diameter × 5.0 mm long pellet	Dry pellet	Wheat whole hard red winter, wheat gluten meal	Lower retention, temperature, and salinity affects dry matter retention rate	N/A	N/A	N/A	N/A	N/A	Obaldo et al. (2002) [134]
	Extruded pellet	Noodle-like 4–5 mm strands	Dry pellet	Wheat flour, wheat gluten	Dry matter loss of the test feeds ranged from 6.3 to 10.6%	Visible behavioral differences among shrimp were apparent immediately after access to the feed	N/A	Shrimp fed krill meal registered maximum weight gain. >86% survival in all treatments	N/A	No difference in FCR or yield among the various treatments	Suresh et al. (2011) [41]
	Extruded pellet	3 mm × 5 mm strand	Dry pellet	Wheat flour	5-min leachate of intact pellet without any krill meal additive-strongest binder	Feed containing krill meal (as low as 1% up to 6%) enhanced ingestion of pellets	N/A	Krill meal effectively enhanced growth (with chemostimulants to enhance palatability)	N/A	N/A	Derby et al. (2016) [43]
	Commercial extruded pellet	Strands with die plate-1.4 mm in diameter	Dry pellet	Wheat starch	N/A	N/A	N/A	Growth rates negatively correlated to the inclusion of dietary pea protein	N/A	N/A	Martínez-Rocha et al. (2012) [108]
	Extruded pellet	3 mm diameter pellet	Dry pellet	Whole wheat	N/A	Diets with FM to SBM replacement showed good feed palatability	Non-GM soy cultivars producing SBM had higher digestibility than white flakes or pressed soy cakes	The diet incorporating ingredient-17 (SBM; de-hulled, roasted, hexane-extracted, and ground) showed the largest weight gain	N/A	N/A	Fang et al. (2016) [112]
	Steam pellet	3 mm pellets	Sinking pellet	Lignosol, agar	Higher dry matter loss in pellet with binder lignosol through micro coating	All pellets were readily consumed by the shrimps	N/A	Weight gain was higher for *Palaemon elegans* than *P. varians* fed diets with lignosol added by microbinding diet		FCR was higher for *P. elegans* compared to *P. varians*	Palma et al. (2008) [44]
	Extruded feed	3 mm diameter pellet	Dry pellet	Aquabind	N/A	Difloxacin was palatable at the 1× treatment level (100 mg/kg of feed)	N/A	Mean weight gains by shrimp receiving difloxacin did not correlate with feed consumption	N/A	FCR were higher in shrimps fed difloxacin-medicated diets	Park et al. (1995) [45]
	Commercial pellet	Bead form with a diameter of 2 mm	Dry pellet	Cod oil, starch solution (3%), squid ink-sac liquid	The melanin coated with starch solution was strongly bound inside the feed	N/A	N/A	Melanin-coated starch solution and melanin coated FO had protection rates of 57% and 67% at Day 7, respectively	N/A	N/A	Thang et al. (2019) [151]
	Extruded pellet	N/A	Dry pellet	Wheat flour, corn starch	N/A	N/A	N/A	Shrimp fed diets of formulated FM showed significantly higher WG and specific growth rate (SGR)	Shrimp fed diets formulated with FM with a significantly higher PER	No significant difference between protein sourced from FM and soy meal	Gil-Nunez et al. (2020) [47]
	Extruded	3 mm diameter pellet	Dry pellet	Whole wheat, corn starch	N/A	N/A	Higher apparent digestibility of dry matter, energy, protein	Increased protein and energy digestibility of an ingredient contributed to higher growth performance	N/A	N/A	Galkanda-Arachchige et al. (2019) [46]
	Steam pellet	1 mm diameter pellet	Dry pellet	CMC	N/A	Feed consumption was higher in the 50% meat & bone meal with garlic supplementation	N/A	SGR were higher in shrimp fed with supplementation of meat and bone meal with garlic compared to meat and bone meal alone	High PER was recorded in feeds supplemented with meat and bone meal with garlic	Highest FCR was recorded in feeds supplemented with 50% meat and bone meal with garlic	Tazikeh et al. (2019) [115]
	Steam pellet	N/A	Dry pellet	CMC	N/A	N/A	Apparent digestibility of feeds & ingredients higher in fish fed the bioprocessed protein	Shrimp fed the bioprocessed protein concentrates showed significantly higher growth performance at 30% FM replacement	Shrimp fed the bioprocessed protein concentrates showed significantly higher PER	Shrimp fed the bioprocessed protein concentrates showed significantly higher feed efficiency (FE)	Moniruzzaman et al. (2019) [114]
Crayfish	Steam sinking pellet	5 mm diameter conglomerated structured	Moist pellet	Carrageenan, CMC, agar, and gelatin	5% binder retained more dry matter compared to 3% binder	N/A	N/A	N/A	N/A	N/A	Ruscoe et al. (2005) [136]
	Extruded pellet	1 cm diameter spaghetti like structure	Dry pellet	CMC	N/A	Some redclaw fed Diet 3 (0% cholesterol and 0.5% lecithin) did not appear to aggressively consume the diet efficiently	N/A	Diet 4 containing menhaden FM, SBM, choline chloride, cod liver oil, and corn oil may satisfy the lecithin and cholesterol requirements	N/A	N/A	Thompson et al. (2003) [13]
	Extruded Pellet	1 cm × 0.1 cm diameter with Spaghetti into cylindrical from	Dry pellet	Pectin, alginate, and chitosan	Pectin diet showed good water stability	N/A	N/A	Pectin diet showed highest wet gain	Pectin diet showed better PER	Chitosan diet showed highest FCR	Volpe et al. (2012) [17]
	Extruded pellet (Stable and unstable pellet)	N/A	Dry pellet	Maize, oat flour	Stable pellets promoted lower leaching rate & faster growth than unstable diets	Marron handled and ingested the intact stable pellets, and ingested unstable pellets for as long as they stayed in the form of a pellet	N/A	Crayfish fed stable diets had higher SGR than the unstable diets and control feed	N/A	N/A	Jussila and Evans (1998) [49]
**Crab**	Steam pellet	Strands pellet with 3–5 mm length	Dry pellet	CMC	N/A	N/A	14.7–17.6 MJ/kg	Crabs grew well when fed diets containing 32–40% protein with either 6 or 12% lipid	20.5–31.1 mg protein/kJ	The FCR, intermolt duration, and total number of days of feeding test diets were not affected by dietary treatments	Catacutan (2002) [50]
	Extruded pellet	4–6 mm length	Dry pellet	Tapioca starch	N/A	Voluntary feed intake in crabs may increase the intake of a low-lipid diet, which were higher at 6% lipid	N/A	Maximum SGR was obtained when the diets were supplemented with 6.57% oil	The highest feed conversion ratio was observed in crabs at 6% lipid feed	Lowest protein efficiency ratio was observed in crabs at 6% lipid feed	Zhao et al. (2015) [117]
	Extruded pellet	4–6 mm length	Dry pellet	Dextrin	N/A	N/A	N/A	Pellet with FO or mixture oil-higher survival	N/A	N/A	Zhao et al. (2016) [51]
	Steam pellet	1.2 mm diameter; 4.0 mm length	Dry pellet	Guar gum	Pellets showed higher water stability after 4 h of immersion	Crabs showed good voluntary feed intake of the feeds (mixed oil refers to vegetable oil and cod liver oil)	Higher apparent digestibility recorded in diets from mixed oil	Crabs fed with mixed oil recorded the same SGR as crabs fed with cod liver oil alone	Similar PER value (1.44 to 1.46) for mixed oil comparable with cod liver oil alone	Similar FCR recorded for crabs fed with mixed oil and cod liver oil alone	Unnikrishnan et al. (2010) [52]
	Steam pellet	~1.2 mm diameter; 4.0 mm length	Dry pellet	Guar gum	Pellets showed higher water stability in all feeds	The crabs fed with CP-20 (20% dietary protein) showed the lowest voluntary feed intake (VFI)	Lower apparent digestibility of protein	The best growth performance, as well as nutrient turn-over, was recorded in crabs fed with 45% crude protein in their diet	The highest PER was obtained by feeding the crabs with CP-20	The FCR was found to decrease with an increasing dietary protein level up to 45% (CP45)	Unnikrishnan and Paulraj (2010) [53]
	Steam pellet	1 × 1 × 0.3 cm jelly cubes	Moist pellet	Agar-agar	N/A	N/A	N/A	Crabs fed diets supplemented with 0 and 2% oil mixture had lower weight gain	N/A	N/A	Sheen and Wu (1999) [118]
	Steam pellet	1 × 1 × 0.3 cm jelly cubes	Moist pellet	Agar-agar	N/A	N/A	N/A	Crabs fed the diets containing 0.5 and 0.79% cholesterol had higher weight gain, whereas 1.12% cholesterol had an adverse effect on mud crab growth	N/A	N/A	Sheen (2000) [119]
	Steam pellet	1 × 1 × 0.3 cm jelly cubes	Moist pellet	Agar-agar	N/A	N/A	N/A	The weight gain of crabs fed diets containing DHA or ARA was higher than those fed the diets without supplemented PUFA	N/A	N/A	Sheen and Wu (2002) [120]
**Lobster**	Steam pellet	5–9 mm cylindrical rod pellet	Moist pellet	Aquabind	Regression analysis showed no significant difference in water stability between pellets	Lobsters fed with pellets from krill meal had greater feed intake than lobsters fed with pellets from FM	Lobsters fed with pellets from FM and krill meal; had greater energy than pellets from fresh items homogenized	SGR in lobsters fed with mussel was higher than other formulated feeds	N/A	N/A	Marchese et al. (2019) [18]
	Steam pellet	String size	Dry pellet	CMC	Leaching of nutritional components accrediting 30 min	Diets were immediately ingested by the lobsters in the first 30 min	Feed with squid meal increase nutritional value & enhances digestive activity	N/A	Supplement of high-quality local fish/squid meal increase protein efficiency	N/A	Perera et al. (2005) [54]

N/A: Not available.

### 4.3. Current Status of Nutritional Research and Developments

Many studies have evaluated adjustments to decapod crustacean feeding formulations by reducing the dependency on FM (protein source) and FO (lipid source). Recent research has explored the use of protein and lipid sources from various sources: terrestrial animal-based materials, plant-based materials, insect meal, food waste, and fishery and aquaculture byproducts [11]. The use of these alternative sources is often evaluated through several reliable indicators such as the voluntary feed intake, feed conversion ratio (FCR), and protein efficiency ratio (PER) in determining the effectiveness of the feed. Feed that uses both FO and FM ingredients has confirmed efficiency in decapod performance in terms of FCR (1.8) and PER (2.8) [47], and, thus, they have been used as a baseline to develop a new feed formulation that uses other protein and lipid sources.

The use of proteins and lipids sourced from plant-based materials is gaining attention nowadays. Plant proteins represent the major dietary protein source used in feed for decapods, after FM and FO. Plant protein sources such as camelina meal, canola meal, and SBM can be used as substitutes for FM without negative effects on growth and feed intake [145]. According to Bae et al. [152], SBM or sesame meal could replace 20% of FM (in a total of 30% of diet inclusion) with PER values of 2.93 and 2.70, respectively, to encourage higher growth performance in juvenile whiteleg shrimp. In a similar study, Yue et al. [153] reported that combinations of SBM and peanut meal can substitute for FM in the diet of whiteleg shrimp from 30% to 20% with PER and FCR ranges of 2.08–2.15 and 1.25–1.30, respectively. On the other hand, the use of SBM as a replacement for FM (60% diet inclusion) in blue swimming crab juveniles indicated that dietary SBM can replace up to 40% of FM in the diet without reducing their growth performance [154]. A grow-out experiment on juvenile Indian prawns (*Fenneropenaeus indicus*) showed that the substitution of dietary FM with SBM solid state fermented with yeast (*Saccharomyces cerevisiae*) can replace up to 50% of dietary FM protein with similar growth performance (PER = 135; FCR = 1.85), feed utilization, and palatability of prawns to the control diet (40% FM) [155]. The total replacement of FM (65%) can be implemented in the diet of juvenile red claw crayfish with a crude protein level reduced to 35% (0% FM and 56.5% SBM) if a combination of plant protein ingredients (SBM, wheat, brewer’s grains with yeast, and milo) is added, with FCR recorded at 5.73 ± 1.2 [156]. Meanwhile, for the replacement of FO with vegetable oil, Zhao et al. [51] found that lipid sources from 8% mixed oil (FO, safflower oil, and perilla seed oil at 1:1:1) with total lipid of 9.6% led to the best performance in juvenile yellow mud crabs.

Poultry byproduct meal is frequently used as a substitute for FM due to its high protein content, relatively low price, and consistent availability. The use of terrestrial animal proteins such as meat and bone meal as replacement for FM has confirmed efficiency in terms of providing decapods with adequate protein [65]. Tazikeh et al. [115] recorded high PER and FCR (1.70 and 1.72, respectively) in whiteleg shrimp fed 50% meat and bone meal with garlic in the diet in comparison to the control diet (45.4% FM). At the same time, fishery and aquaculture byproducts and insect meal showed high potential as protein sources substituting for FM through the biotransformation and bioconversion of raw waste materials [11]. Hence, there has been a compelling need to increase the dietary use of these alternative protein sources, given their ability to supply decapods with the required protein levels.

However, major setbacks associated with these alternative protein resources include the lack of attractants and a reduced palatability factor [157]. Compared to feed derived from aquatic animals (i.e., FM, shrimp meal, and squid meal), the lack of attractants may result in poor ingestion of feed, thus reducing feed intake and retarding growth in the cultured species [158]. As such, SBM, without the supplementation of feed additives, was not suitable as a major protein source in the diet of juvenile giant freshwater prawn [159]. In addition, experiments on juvenile red claw crayfish showed that the replacement of FM with SBM, rapeseed meal, or peanut meal had an adverse influence on the growth of the crayfish [160]. At the same time, some of the rendered animal protein meals, such as blood meal, hydrolyzed feather meal, or meat and bone meal often have deficiencies in essential amino acids and thus are lacking in terms of attractability, especially when used as the main source of protein in diets [46]. There is the need for additional feed attractants for the formulation of pellets that use plant protein to ensure maximum attractability and palatability for decapods. The inclusion of krill meal as an attractant in the feed formulation increased the acceptance of SBM, the primary protein source, by homarid American lobsters [161] and whiteleg shrimp [162].

Furthermore, the development of formulated feed used protein hydrolysates produced from purified protein sources through the addition of proteolytic enzymes. The protein hydrolysates contain mixtures of peptides and free amino acids that help to attract decapods. These ingredients are rich in amino acids that act as chemical signal compounds detectable by the chemosensory systems of decapods [41]. Feeding stimulation of purple mud crabs (*Scylla tranquebarica*) using 1 mL sugarcane juice of different concentrations in the diet of the mud crabs revealed that the crabs approached the test feeds at 50% and 100% concentrations of sugarcane juice [163]. Overall, stimulants in the decapod feeds helped to minimize the leaching of nutrients and waste through the manipulation of feeding behavior.

## 5. Conclusions

The importance of good pellet physical characteristics in decapod feeding cannot be overemphasized in order to ensure that decapods meet their nutrient needs. The current development of decapod formulated feeds is focused on the juvenile stage. However, the unique feeding behaviors of adult decapods (slow feeding, bottom dwelling, and aggression when handling feed) are major challenges to developing a high-quality pellet for adult decapods. A high-quality pellet not only depends on the binding agent, but also on the attractants that enhance palatability, as well as the correct proportion of nutrients to boost decapod performance. However, most studies published on decapod nutrition lack data on the physical qualities of the feed. Thus, it is difficult to establish a standard feed formulation that focuses on the physical pellet properties.

## 6. Future Research Directions

After reviewing the decapod pellet forms, taking into account decapod growth performance, feeding behavior, and optimum diet composition, further research is needed in the following areas: (1) future feed developments must adhere to the physical feed quality criteria; (2) special attention should be paid to the proper use of feed attractants to increase palatability and acceptability, while at the same time reducing the residence time of the pellets in the water so that decapods can fully utilize the nutrients provided; (3) pellets that offer a nutritionally balanced diet that is stable and digestible must be developed; and (4) further research is needed to explore the potential of the pelleted feed to be commenced at the adult stage so that the decapod can fully reap the benefits of the nutrients present in the pellets.

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
