# Peer review of "Use of Pelleted Diets in Commercially Farmed Decapods during Juvenile Stages: A Review"

_animals, 2021, doi:10.3390/ani11061761_

Round 1

Reviewer 1 Report

The manuscript is proposing a revision on food, feeding and diets in crustaceans. These are a large scope to review. A scientific review paper need provide a comprehensive and deep foundation on a topic, explain the current state of knowledge, identify gaps in existing studies for potential future research and highlight the main methodologies and research techniques. Moreover, is highly desirable that authors propose some further directions to research of topic under review. Unfortunately, this manuscript is farway to it. It is almost impossible meet all these assumnptions with a scope too large. Only one these (food, feeding or diets) would be a topic to one extensive scientifc review. In fact, excluding tables and refrences, few discussion on topics are provided in text. Moreover, the major topics has basic approach, becuse, again, each factor has many aspects to be deeply discussed in a scientific review. 
For example, to relationship between natural food and exogenous feeds:
How much crustaceans take of natural food in different production systems? What the preferences of crustaceans to grazing natural food communities? How differences between species, groups, feeding habits, to point ou a few aspects, can influence feeding? What  are the main biotics and abiotics factors that influence ingestion of food in crustaceans? How these influnce they? 
An several other topics can be provided... In the present status, the manuscript can be published as a book chapter or technical paper, but is not adequate to be published as scientific review in Animals. 

Reviewer 2 Report

I added my comments to the attached file

Reviewer 3 Report

The authors have reviewed the pellet forms and nutrient composition of diets in Crustaceans (CR). This is an interesting topic that has filled the gap in our knowledge of CR nutrition. The manuscript (MS) was well-written, and most of the parts were easy to understand. They covered most of the topics that were needed. However, some issues have compromised the quality of this and need to be addressed.

Major comments

  • First, the English of the manuscript needs to be improved with a native English speaker. Some parts needed to be revised, which I mentioned to some of them.
  • I did not see any section for “feeding behavior” and should be added to the MS.
  • Abstract and Simple Summary have to be massively revised as they do not present the contents of this MS well.
  • The authors did not investigate any interaction in this MS, which does not make sense for this MS. It is better to remove this term throughout the MS. I suggest using just a title like A review in used pellet forms, diet composition and feeding behaviors in crustacean: an update from 21st century
  • I suggest adding the carbohydrate section to the text to complete the puzzle: protein, lipid, and carbohydrate.”
  • I suggest authors checking each reference precisely to ensure whether they are matched with the concepts or not. In this way, references and some sentences need to be revised; some works in fish have to be removed from the text. In any aspect of nutrition, enough references are available for CR. As this MS is for CR, all parts have to be cited and discussed around CR and not fish. Another point is that, in some parts, they generalized some concepts that are just true for some species. Doing these things will help to narrow your MS to only CR.
  • It is better to focus on literature from recent years as we have learned so much about this topic in the last few years.

Minor comments:

 Simple Summary:
Please check what the difference is between “food” and “feed.” I think  “feed” is an appropriate word as an animal eats it. Please check this throughout the MS.

This section needs to be revised; it is not clear what this MS is precisely about.

  • Line 17, change to “crustaceans production.”

Abstract

  • Line 18, please move this part somewhere else. It needs references and, thus, is not a good start for the abstract.
  • Line 19, please revise this part; when you say “interaction,” it means you should investigate all relationships between them, which is impossible. Please revise this point throughout the MS.

  • Introduction:
    • Line 30-31; please be consistent with CR, and if this part is not for CR, remove it. As this MS is for this taxon, no need to report and cite fish works.
    • Line 31-34, please remember this point and revise the MS again: natural feeds are optimum, and we are making artificial diets as we can not keep using natural resources, and the sustainable way is formulating artificial diets. But regarding nutrient contents, I can say natural feeds even better than artificial diets.
    • Line 37, references are old here, and probably this sentence is not true now for these species. Please update this part or revise it.
    • Line 38-40, is not true; please revise it; hips of work are available.
    • Line 41-43, please revise this part
    • Line 44-59 You need to relocate this part somewhere earlier. Also, as I mentioned earlier, it is better just to focus on studies and works on CR and not fish.
    • Line 44-46, this part needs to be revised, is not clear. Fish species that are used for fishmeal is not usually consumable for human, which kind of competition do you mean? Please explain more about this part.
    • Line 48, change to “crustaceans feeding formulation” I think you meant fish meal. Please revise the MS from this point and make sure you used “fish meal” and fish in the correct form and place.
    • Line 49- it should be “meat and bone meal.”
    • Line 52, change it to CR; this reference is for fish
    • Line 57, this reference is for fish, revise this part
    • It is not clear what do you mean by “food types”. Please define this term clearly somewhere upper.
    • Line 74, feed type, is true; all food types need to be changed to “feed types”. With checking Table 1, I suggest changing “feed type” to “pellet forms”. It will be much more understandable for readers. Please change this term throughout the MS.
    • Also, in Table 1, I suggest adding some references which are about “live feed and natural diets”. It can give an idea to readers what is the difference between these feeds with artificial diets.
    • Line 76-77, change to meanwhile most of the studies for lobster and prawn (and some on crab), focused on the adult or broodstock stages.
    • Line 77-79, move this part to somewhere earlier in the introduction when you mentioned for the first time to “feed type.”
    • line 79, remove “from each other”.
    • Line 82, “water activity,” please explain what this is and then continue with lines 87-88.
    • Also, please define the size that you considered as Juvenile, Subadult, Adult, Broodstock, Immature / Mature,
    • Line 120-122, please revise this part.
    • Line 125, revise this part
    • Line 130-131, please cite a work in CR and not fish.
    • Line 143-145, we have some commercial wet feeds and I think this part needed to be revised.
    • Line 154-155, please revise this part. Did you mean CR can find feed faster if the diet is more palatable?
    • Line 168-170, relocate this part as is the answer to my above question as well.
    • Line 183, please add references for CR here.
    • Line 168, please recheck these references, and also add references in CR here
    • Line 201-203, please revise this part.
    • Line 205-206. which CR are using this diet? Please give an example here and if you can bring references as well.
    • Line 210, please explain more about this part, is not clear enough.
    • Line 218, change feed intake to “feed consumption”. The reason is that feed intake has another concept that is not matched for here.
    • Line 217, change to “A higher water stability in pellets…
    • Line 222, please change with references in CR
    • Line 227-228, change to CR.
    • Line 231, change to CR
    • Line 238-240, please revise this part and just mention and cite references in CR.
    • Line 226, this section is quite repetitive as you already mention these factors in the last sections, and I suggest removing this part.
    • Line 249, change to “nutrient contents of diets
    • Line 250-258, please revise this part massively; also, please use some better connections to link the sentences. You started the paragraph with nutrients and then reproduction and survival and finally protein and lipid. There is no connection between them.
    • Line 282-284, please revise this part; it is not clear
    • Line 308-310, I suggest adding more information from other species of CT. These two references are in just crab.
    • Line 325, here and elsewhere, try to use a better term: n-3 LCPUFA.
    • Line 325-338, please just cite CR references and update this section.
    • Line 323, this MS is not about all animals and is just for CR. Therefore, throughout the MS, just focus on CR.
    • Line 332-334, please cite more relative works here and also, ARA can promote growth as well, and this sentence needs to be revised.
    • Line 337: I suggest adding the carbohydrate section to complete this puzzle.
    • I suggest adding a section for future researches and outlines.
    • Line 343, you did not discuss any interaction in this MS; therefore, “interaction” should be removed from the title, here, abstract, and elsewhere.

Best regards

I hope I have contributed to improving this MS.

Round 2

Reviewer 1 Report

The authors done a significant improvement in writing and reduced the focus of the manuscript. However, the theme proposed remain very large. Approach use of pelleted diets in all shellfish species is very hard. Thus, some aspects detected in the previous version remain, such as a superficial approach in some topics. Several comments are present as annotations in the pdf file.

Reviewer 3 Report

Although this MS has been improved, some more works are required to get close to the final version.

Major revision:

  • I can see many parts are repetitive, and I suggest authors massively revise the MS again from this point. By doing this, many parts will be summarized.
  • In many parts, authors just mention crab, shrimp,… and this is not enough. Please be clear with the name of the species. For the first time, mention the common name along with the scientific name, and the rest just report the common name.

Minor revision:

  • Line 13, change to: to develop a high-quality formulated diet in aquaculture….
  • Line 16, change to: for the sustainability of crustacean aquaculture.
  • Line 20, delete “together…formulation.
  • Line 24, change to “feed.”
  • Line 25, change from “To close this knowledge gap” to broaden the horizon of this area..
  • Line 26, change identify to suggest
  • Line 29, delete “and with no restriction to only certain species.”
  • Line 36, change good to “high quality.”
  • Line 37, here and elsewhere, please check this phrase accurately; pellets mean diets, so no need to say “pellet diet”. You need some time to use a “pelleted diet” that I think is not applicable for this MS. Therefore, just use pellet.
  • Line 41, change to “By manipulating the levels of protein and lipid, a formulated feed can be expected to provide optimum growth and reproduction in crustacean.
  • Line 45, fish meal is a source of protein and not lipid. You can say fish oil for lipid
  • Line 47, delete “in providing the crustacean species with good protein levels”. It is also suggested to exchange this reference with a more recent one https://doi.org/10.1111/are.14416.
  • Line 51, change from “the use of protein source from the terrestrial animals and plant origin” to: these alternative protein resources.”
  • Line 61-63, I suggest deleting this part
  • Line 108, change to is one that is formulated….
  • Line 109, change to “to achieve the optimum growth performance and cost
  • line 151-152, change to “promoting high growth and feed efficiency owing to their soft texture and palatability.
  • Line 712, change from “good” to “high
  • Line 192, revise this part
  • Line 203-206, I suggest deleting this. Also, most of the formulation is based on dry weight, and this part does not make sense. Please revise this part, or you can even delete it.
  • 215, “Good, palatable diet helps in good digestion of crustacean” this is not true; please delete it.
  • Line 218-221, delete this part as information was wrongly presented.
  • Line 207-236, please review it again and make sure you say just one time that palatability and attractability improve growth. Many parts, therefore, can be removed.
  • Line 238-264, please make you do not repeat some information. For example, please mention one time that binders minimize nutrient leaching.
  • I suggest you bring section 4.3 before 4.2, and it is better you mention binder after water stability.
  • Line 275, no one use floating pellets for crustacean in the non-larval stage; please recheck the MS from this point. If you think it is common, please bring valid references for that.
  • Line 281 and elsewhere, make sure you use abbreviations if you used the term more than three times in the MS. Otherwise, just only report the whole phrase, and no need to abbreviate them.
  • Line 315, please revise this part and make sure you differentiate the larval stage from other stages. Still, I think using a floating pellet for crustacean in the non-larvae stage, cannot be an appropriate way of feeding.
  • Line 397 and 403, please change one of these “on the other hand”
  • Line 455-467, I suggest removing this part and just start with 6.1 protein
  • Line 471, remove “of crustacean.”
  • Line 471-672, too general; I suggest removing this part.
  • Line 511, for the first time, use a common name along with a scientific name, and for the rest, just a common name.
  • Also, throughout the MS, please use the scientific name for all mentioned details. Just using shrimp or crab is not enough. From this point, the MS should be massively revised.
  • Line 546- 550, please remove this part as it is too general. Also, for the nutrients part, I suggest summarizing the paragraphs by deleting some parts that are too general.
  • Line 577, change to “revealed that individuals fed….”
  • Line 578-580, are you sure about it? What about marine sources for providing EPA and DHA!!
  • Line 605, dietary level of?
  • In the nutrients section, please make it more numeric and be clear how much is when you say high or low contents of protein, lipid, and carbohydrate.
  • Also, please bear in mind that carbohydrate is not just for energy and has some other benefits. I suggest you adding this part as a short paragraph.
  • Line 612-614 here and elsewhere, please make sure again to not repeat the information. Please read the revised MS again and delete the repetitive parts.
  • Line 635-636, please revise it is not clear
  • Line 637, you already mention this in the abstract; please remove one of them.
  • Line 636-641, please revise it; I cannot see any connection here
  • Line 643 change from good to high.
  • Line 641, “The complexity of crustacean nutrition requirement,” what do you mean by this
  • Line 647-649, this part is suitable for the next section.
  • Line 647 and elsewhere, you can “suggest and recommend” and “must” is not a correct verb. Revise the MS from this point.

Best regards

I hope I have contributed to improving this MS.

Round 3

Reviewer 1 Report

Dear authors,

No additional comments are needed. Congratulations.

Reviewer 3 Report

Authors have improved the MS and before getting close to final acceptance, some minor things can be fixed.

  • I can still see you did not use the common name and scientific names appropriately. For the first time of the MS, use common name plus scientific names, and the rest use the common name. Please avoid using like “mangrove crab, Scylla sp”, blue swimming crabs Portunus sp. etc.
  • Line 15, I understand you did these changes according to other reviewers' comments, but is it the aim of this review “to develop a high-quality formulated diet in aquaculture”?. You just reviewed and did not do any original research and experiment that you want to develop.
  • Line 33, again, I think you focused on adults as well in this review. If yes, I suggest deleting the juvenile stage.
  • Line 66, what do you mean by “compound diet.”
  • Line 80 and elsewhere, please check you mentioned to complete terms for the first time in MS and then use abbreviations
  • Line 244, please bring references,
  • Line 249, make sure you define the abbreviations for the first time.
  • Line 324, change to Crustacean Hyperglycemic Hormone.
  • Line 356-259, as I said in the past versions, make sure you already checked what you cite. So if 10% is the maximum for CHO, it would have 35-40% proteins and 5-10% ash and would be 30% lipid!!!!!!!!!!!!!!!!! Which is wrong.
  • Line 362-270, I suggest deleting this part as it is too much for CHO, and also, this paragraph is not relevant enough.
  • Line 588-616 and 621-644; I think other reviewers wanted to have these here. However, I am not sure how it is fixed to your topic. Please update this part with the level of FM in the control diet as just 40% replacement does not say anything without knowing the FM level in control.
